# UVR8 disrupts stabilisation of PIF5 by COP1 to inhibit plant stem elongation in sunlight

Ashutosh Sharma [1], Bhavana Sharma[1], Scott Hayes [2], Konstantin Kerner[3], Ute Hoecker[3], Gareth I. Jenkins[4] & Keara A. Franklin[1]*

Alterations in light quality significantly affect plant growth and development. In canopy shade, phytochrome photoreceptors perceive reduced ratios of red to far-red light (R:FR) and initiate stem elongation to enable plants to overtop competitors. This shade avoidance response is achieved via the stabilisation and activation of PHYTOCHROME INTERACTING FACTORs (PIFs) which elevate auxin biosynthesis. UV-B inhibits shade avoidance by reducing the abundance and activity of PIFs, yet the molecular mechanisms controlling PIF abundance in UV-B are unknown. Here we show that the UV-B photoreceptor UVR8 promotes rapid PIF5 degradation via the ubiquitin-proteasome system in a response requiring the N terminus of PIF5. *In planta* interactions between UVR8 and PIF5 are not observed. We further demonstrate that PIF5 interacts with the E3 ligase COP1, promoting PIF5 stabilisation in light-grown plants. Binding of UVR8 to COP1 in UV-B disrupts this stabilisation, providing a mechanism to rapidly lower PIF5 abundance in sunlight.

[1] School of Biological Sciences, Life Sciences Building, University of Bristol, Bristol BS8 1TQ, UK. [2] Centro Nacional de Biotecnología (CNB-CSIC), Calle Darwin 3, Madrid 28049, Spain. [3] Botanical Institute and Cluster of Excellence on Plant Sciences (CEPLAS), Biocenter, University of Cologne, Cologne, Germany. [4] Institute of Molecular, Cell and Systems Biology, College of Medical, Veterinary and Life Sciences, University of Glasgow, Glasgow G12 8QQ, UK. *email: kerry.franklin@bristol.ac.uk

In shade intolerant plants, the detection of neighbouring vegetation triggers a suite of developmental responses, termed shade avoidance. These include the rapid elongation of stems to elevate leaves towards sunlight[1]. Early neighbour detection includes the touching of leaf tips and reduction in the red to far-red ratio (R:FR) of reflected light[2]. Following canopy closure, additional reductions in blue:green ratio and UV-B signal true shade[3]. Reductions in R:FR inactivate phytochrome photoreceptors, promoting the stabilisation and activation of PIF transcription factors. PIFs 4, 5 and 7 perform a key role in the regulation of shade avoidance, by binding to and activating auxin biosynthesis genes[4,5].

On emerging from a canopy, UV-B provides an unambiguous sunlight signal which inhibits further shade avoidance[6,7]. UV-B is perceived by dimers of the photoreceptor UV RESISTANCE LOCUS 8 (UVR8). These monomerise following UV-B absorption and interact with the CONSTITUTIVELY PHOTOMORPHOGENIC 1 (COP1)/SUPPRESSOR OF PHYA-105 (SPA) complex, promoting the stabilisation and expression of ELONGATED HYPOCOTYL 5 (HY5) and HY5 HOMOLOG (HYH) which, in turn, drive UV-B-signalling[8]. UV-B-mediated inhibition of shade avoidance involves multiple mechanisms which serve to inhibit PIF activity, including the stabilisation of DELLA and HY5 proteins[6]. These form non-DNA-binding heterodimers with PIFs and compete with PIFs for target promoters, respectively[9,10]. UV-B has additionally been shown to promote the rapid degradation of both PIF4 and PIF5 in low R:FR[6].

The molecular mechanisms controlling PIF abundance in UV-B are unknown. Here we show that activation of the UVR8 photoreceptor promotes rapid PIF5 degradation via the ubiquitin-proteasome system and requires the N terminus of PIF5. We further demonstrate that PIF5 physically interacts with COP1 in de-etiolated seedlings. This interaction stabilises PIF5 in low R:FR, consistent with observations in dark-grown plants[11,12]. PIF5 abundance is enhanced in uvr8 mutants in UV-B and decreased in cop1 mutants, suggesting that UVR8 binding to COP1 in UV-B acts to destabilise PIF5, rapidly inhibiting shade avoidance once sunlight has been reached.

## Results

**Active UVR8 inhibits PIF5-mediated hypocotyl elongation.** UV-B can suppress residual shade avoidance responses in pif4, pif5 and pif7 single and higher order mutants (Supplementary Fig. 1). These data suggest that the inhibition of shade avoidance by UV-B involves the suppression of PIF4, PIF5 and PIF7 activities. To examine the role of UVR8 in UV-B-mediated PIF5 degradation, we generated transgenic lines expressing 35S:PIF5-HA in Arabidopsis WT(Ler) and the uvr8-1 mutant background. PIF5 increases hypocotyl length so phenotypes from homozygous over-expressing lines in each background were compared to non-transgenic lines. LerPIF5Ox 5-7, LerPIF5Ox 8–13 and uvr8-1PIF5ox 2-1 showed a long-hypocotyl when compared to Ler and uvr8-1 in continuous white light, whereas uvr8-1PIF5ox 1-3 resembled uvr8-1 controls (Supplementary Fig. 2a). Low dose UV-B strongly inhibited hypocotyl length in LerPIF5Ox 5-7 and LerPIF5Ox 8-13 but not in uvr8-1PIF5Ox 1-3 and uvr8-1PIF5Ox 2-1, confirming the role of UVR8 in this response (Supplementary Fig. 2a). Immunoblot analysis of PIF5 levels showed that hypocotyl elongation phenotypes were proportional to the level of PIF5 protein and confirmed previous observations of UV-B-mediated PIF5 degradation[6] (Supplementary Fig. 2b). The lines, Ler PIF5Ox 5-7 and uvr8-1PIF5Ox 2-1, (hereafter LerPIF5Ox and uvr8-1PIF5Ox), showed a similar PIF5 level and phenotype, so were selected for further study.

In high R:FR, LerPIF5Ox and uvr8-1PIF5Ox showed a long-hypocotyl phenotype when compared to Ler and uvr8-1 controls. Supplementary UV-B (+UV-B) inhibited this elongation in a UVR8-dependent manner (Fig. 1a). Low R:FR treatment promoted hypocotyl elongation in all genotypes and this phenotype was exaggerated in PIF5 over-expressing lines (Fig. 1b). Supplementary UV-B strongly suppressed hypocotyl length in a UVR8-dependent manner, although a UVR8-independent component to this response was also observed (Fig. 1b). Similar trends were observed in 16 h light/8 h dark cycles (Supplementary Fig. 3).

**UV-B perceived by UVR8 rapidly decreases PIF5 abundance.** Previous studies have shown that UV-B reduces PIF5 protein abundance within 2 h[6]. To understand both the kinetics and role of UVR8 in this response, detailed time-course immunoblots were performed using LerPIF5Ox and uvr8-1PIF5Ox lines. Seedlings were grown for 10 days in 16 h light/ 8 h dark photoperiods and samples harvested at predawn and various timepoints following exposure to either high or low R:FR ± UV-B. UGPase was used as a loading control to quantify relative PIF5 levels. Transfer from dark to light (high R:FR) rapidly reduced PIF5 abundance to 50% within 240 min, consistent with phytochrome-mediated degradation[13]. Supplementary UV-B enhanced the rate of PIF5 degradation but the same final level of PIF5 was reached in UV-B-treated and -untreated samples (Fig. 2a, b). Low R:FR increased PIF5 protein abundance, in accordance with previously published observations[13]. UV-B rapidly decreased PIF5 abundance in low R:FR, reducing PIF5 levels to almost 50% of untreated controls. (Fig. 2c, d). To investigate whether UVR8 is involved in UV-B-mediated PIF5 degradation, time-course immunoblots were also performed in uvr8-1PIF5Ox plants. In both high R:FR and low R:FR, supplementary UV-B had no effect on PIF5 abundance, confirming the role of UVR8 in mediating this response (Fig. 3a–d).

**UV-B-mediated decreases in PIF5 involve protein degradation.** We next investigated whether UV-B-mediated reductions in PIF5 resulted, at least in part, from an increase in PIF5 transcript turnover. We quantified the transcript abundance of PIF5 in 10-day-old plants transferred to high R:FR and low R:FR ± UV-B. No UV-B-induced changes were recorded in high or low R:FR at earlier timepoints (0-40 min) in either LerPIF5ox or uvr8-1PIF5ox lines (Supplementary Fig. 4). In high R:FR, reduced PIF5 transcript was observed in UV-B-treated plants relative to untreated controls at timepoints >80 min (Supplementary Fig. 4a). Parallel analyses using primers designed against HA showed similar results (Supplementary Fig. 5a). As UV-B reduced transcript abundance of constitutively expressed PIF5-HA, it is likely that the decrease represents UV-B-enhanced transcript degradation. This response was not observed in uvr8-1, suggesting that it is mediated by UVR8 (Supplementary Figs. 4b, 5b). Interestingly, no UV-B-mediated reduction of PIF5 transcript was observed in low R:FR in either wild-type or uvr8-1 backgrounds (Supplementary Figs. 4c, d, 5c, d). It can therefore be concluded that the UV-B-mediated decreases in PIF5 shown in Fig. 2, are likely to result from increased protein degradation.

**UV-B degrades PIF5 via the ubiquitin-proteasome system.** PIF proteins are regulated by light and their phosphorylation and/or degradation in light controls downstream signalling. In high R:FR, PIF5 protein is degraded by the ubiquitin-proteasome system in a mechanism requiring its N-terminal active phytochrome B binding (APB) domain[13]. We therefore investigated whether a similar system was involved in UV-B-mediated PIF5 degradation. LerPIF5Ox seedlings were grown for 10 days in 16 h light/8 h dark

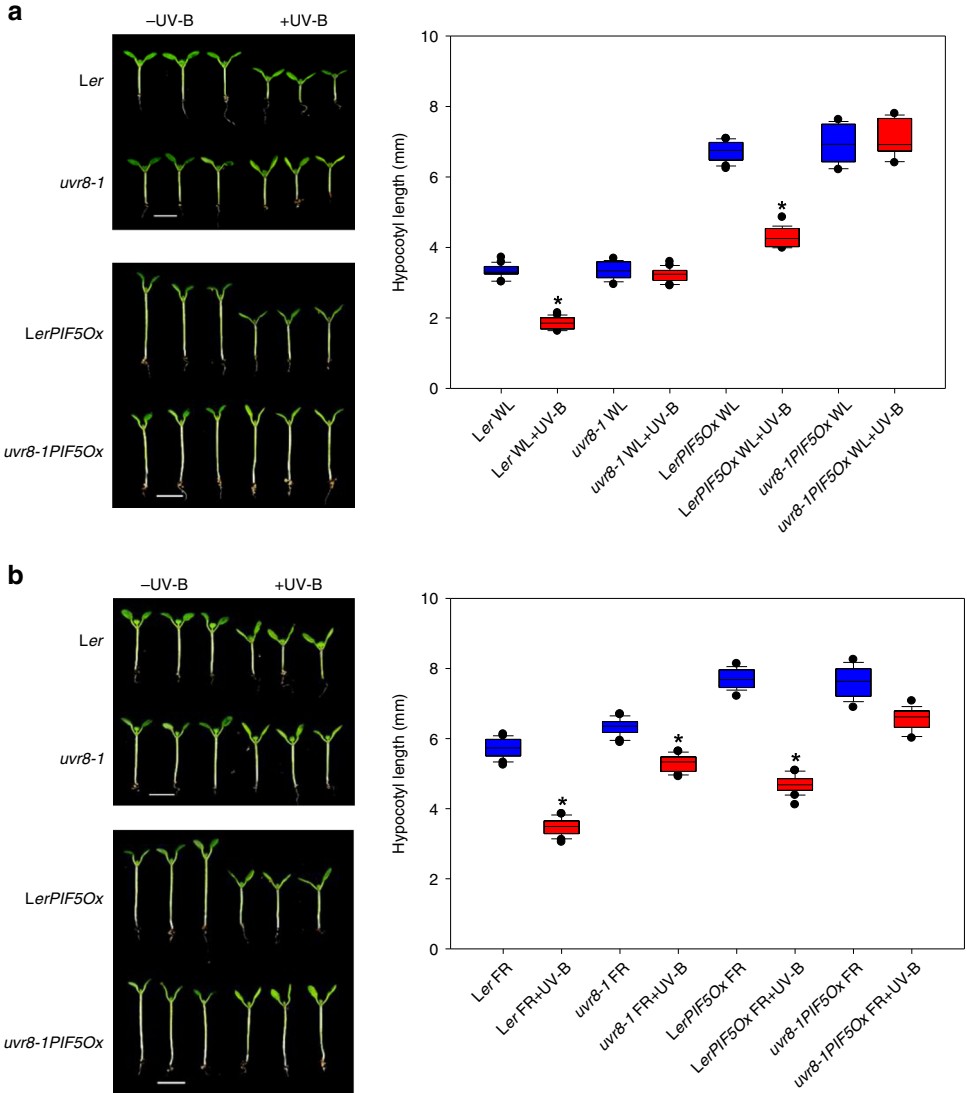

**Fig. 1** UV-B perceived by UVR8 inhibits PIF5-mediated hypocotyl elongation. Representative seedling images and hypocotyl length measurements of L*er*, *uvr8-1*, L*erPIF5Ox* and *uvr8-1PIF5Ox* seedlings grown for 3 days in continuous high R:FR light before transfer to (**a**) high R:FR (WL) or (**b**) low R:FR (FR) for 4 days ± UV-B. Boxes represent 25th to 75th percentile. Bars show the median and whiskers represent the 10th and 90th percentile outlines. *Significant differences when compared to controls without UV-B treatment (Tukey's HSD, P < 0.01, n ≥ 14). Scale bar = 4 mm. Source data are provided as a source data file

photoperiods and treated with the proteasome inhibitor MG132 for 16 h before transfer to low R:FR ± UV-B at dawn. Immuno-blot analysis of PIF5-HA abundance showed that MG132 pre-vented UV-B-mediated PIF5 degradation (Fig. 4a). Blots were also probed with an anti-ubiquitin antibody to confirm that the MG132 had fully imbibed into the tissue and increased the abundance of ubiquitinated proteins (Fig. 4b). Western blot analysis of immunoprecipitated PIF5 extracted from UV-B-treated L*erPIF5Ox* seedlings showed a ladder of ubiquitinated proteins, the intensity of which was enhanced following protea-some inhibitor treatment (Fig. 4c). Together, these results suggest that PIF5 protein degradation in UV-B is mediated by the proteasome-system.

Phytochrome B has been shown to interact with the APB domain of PIF5 in vitro and promote its degradation in red light[14,15]. To investigate the role of the APB domain in UV-B-mediated PIF5 degradation, *35S:ΔNPIF5-HA* transgenic plants lacking the first 68 amino acids, including the APB domain of PIF5[13] were analysed. Control (*35S:PIF5-HA*) and *35S:ΔNPIF5-*

*HA* plants were grown in 16 h light/8 h dark cycles and transferred to high and low R:FR ± UV-B at dawn. *35S:PIF5-HA* immunoblots confirmed previous observations in the Col-0 background. PIF5-HA protein was stabilised in low R:FR and degraded following a 2 h UV-B treatment (Fig. 4d, e; Hayes et al.[6]). In contrast, *35S:ΔNPIF5-HA* plants displayed constitu-tively stable PIF5-HA protein[13] which was unaffected by low R:FR, and UV-B treatments (Fig. 4f, g). These data suggest that the N-terminus of PIF5 is required for UV-B-mediated degradation.

**UVR8 and PIF5 do not interact *in-planta*.** We previously reported a lack of physical interaction between UVR8 and PIF4/PIF5/PIF7 in vitro using the yeast two hybrid system[6] and next wanted to test whether PIF5 interacts with UVR8 in planta, so performed co-immunoprecipitation (Co-IP) assays using *LerPI-F5Ox* plants. PIF5-HA was immunoprecipitated using anti-HA-beads. Clear immunoprecipitation of PIF5 was observed, but no UVR8 could be detected in IP samples, despite detection in the

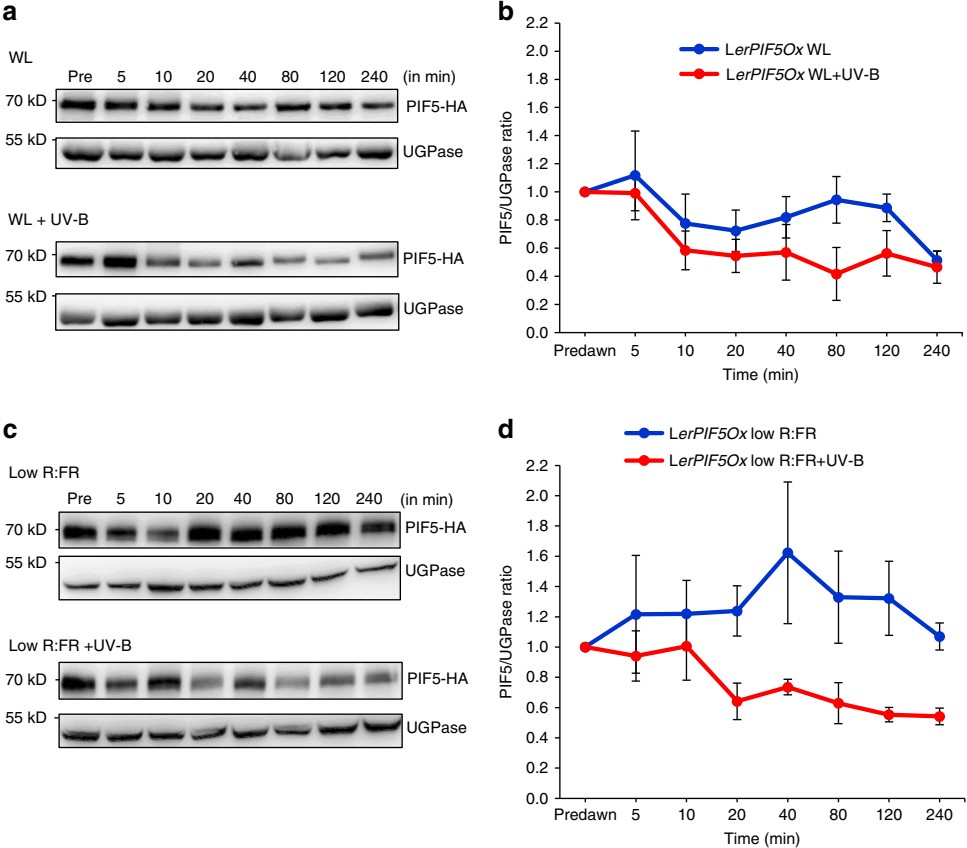

**Fig. 2** UV-B rapidly decreases *PIF5* protein abundance in high and low R:FR. **a** Western blots of *PIF5-HA* and UGP*ase* in L*er*PIF5*Ox* seedlings grown for 10 days in 16 h light/ 8 h dark cycles before transfer at dawn to high R:FR ± UV-B. **b** Quantification of PIF5 protein in three independent biological repeats. **c** Western blots of *PIF5-HA* and UGP*ase* in L*er*PIF5*Ox* seedlings grown for 10 days in 16 h light/8 h dark cycles before transfer at dawn to low R:FR ± UV-B. **d** Quantification of PIF5 protein in three independent biological repeats. Bars represent s.e.m. Source data are provided as a source data file

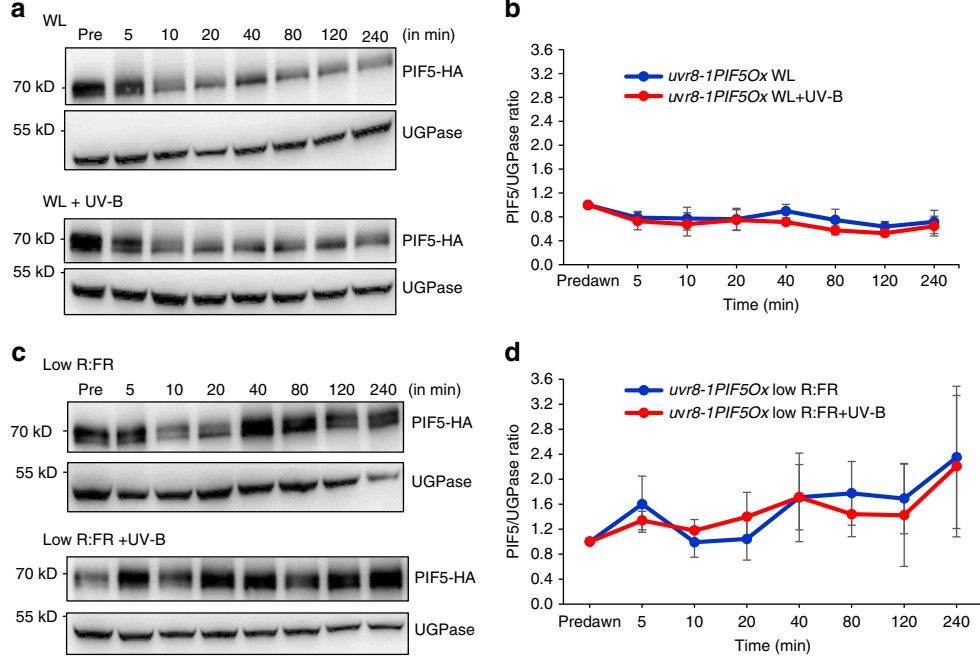

**Fig. 3** UVR8 controls UV-B-mediated decreases in *PIF5* protein abundance in high R:FR and low R:FR. **a** Western blots of PIF5-HA and UGPase in *uvr8-1PIF5Ox* seedlings grown for 10 days in 16 h light/8 h dark cycles before transfer at dawn to high R:FR ± UV-B. **b** Quantification of PIF5 protein in three independent biological repeats. **c** Western blots of PIF5-HA and UGPase in *uvr8-1PIF5Ox* seedlings grown for 10 days in 16 h light/8 h dark cycles before transfer at dawn to low R:FR ± UV-B. **d** Quantification of PIF5 protein in three independent biological repeats. Bars represent s.e.m. Source data are provided as a source data file

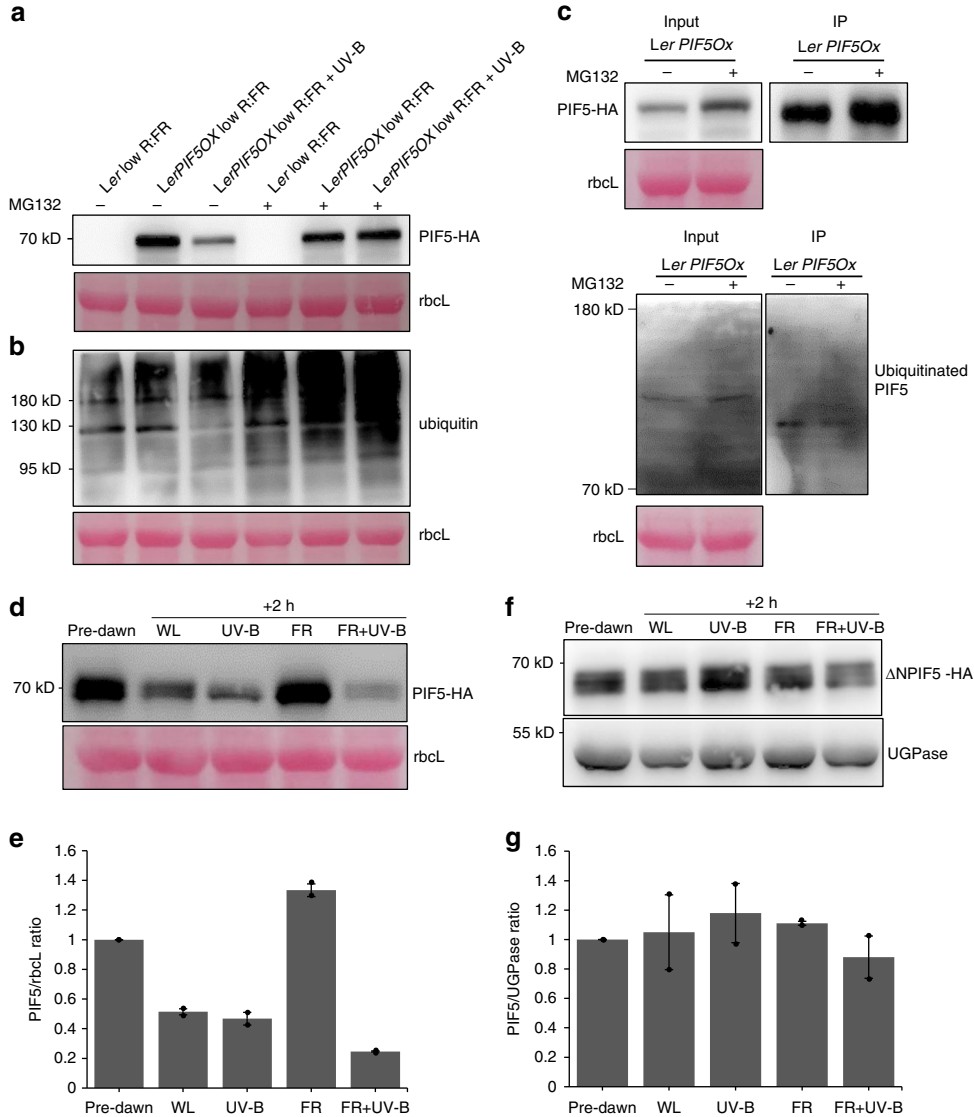

**Fig. 4** UV-B-mediated PIF5 degradation occurs via the proteasome system and requires the APB domain of PIF5. **a** MG132 inhibits PIF5 protein degradation in L*er*PIF5Ox. Plants were grown for 10 days in 16 h light/8 h dark cycles before being transferred to ½ strength MS liquid medium containing 0.1% DMSO ± 50 uM MG132 for 16 h. Plants were transferred at dawn to low R:FR ± UV-B for 40 min. PIF5-HA was detected with an anti-HA antibody. L*er* was used as negative control and ponceau staining of the Rubisco large subunit (rbcL) was used as loading control. **b** Western blot of protein samples from (**a**) probed with an anti-ubiquitin antibody. **c** Co-IP assay showing PIF5 ubiquitination. Seedlings were grown as in (**a**). Total protein extracts were immunoprecipitated from low FR + UV-B 30 min treated samples with anti-HA beads and immunoblots probed with anti-HA or anti-Ubiquitin antibodies. Ponceau stained Rubisco large subunit (rbcL) was used as a loading control. **d** Western blot of PIF5 protein abundance in *35S:PIF5-HA* plants. Seedlings were grown for 10 days in16 h light/8 h dark cycles before transfer at dawn to high R:FR ± UV-B for 2 h. PIF5 was detected with an anti-HA antibody. UGPase was used as loading control. **e** Quantification of PIF5/rbcL ratio in two biological repeats of (**d**). Bars represent s.e.m. **f** Western blot of PIF5 protein abundance in *35S: ΔN.PIF5* lines containing a deletion of the first 68 amino acids of the PIF5 protein. Blots were performed as in (**c**). **g** Quantification of ΔN.PIF5/UGPase ratio in two biological repeats of (**f**). Bars represent s.e.m. Source data are provided as a source data file

input controls (Fig. 5a). PHYB was used as a positive control for the immunoprecipitation of PIF5 complexes and was clearly detected in IP samples (Fig. 5a). Interestingly, PHYB was observed as two bands of different molecular weights. Their identity was confirmed by comparison of input controls with a *phyB* mutant in which neither band was detected (Supplementary Fig. 6a). Immunoprecipitated PHYB was a lower molecular weight, suggesting that phyB-PIF5 interaction results in modification of the PHYB protein (Fig. 5a).

**UVR8 disrupts stabilisation of PIF5 by COP1.** The E3 ligase CONSTITUTIVELY PHOTOMORPHOGENIC 1 (COP1) is a

central repressor of photomorphogenesis and accumulates in the nucleus in shaded conditions[16]. COP1 interacts with UVR8 to promote UV-B signalling[8]. Recently, a noncanonical role of COP1 was identified in PIF3 signalling. Binding of PIF3 to the COP/SPA complex was shown to promote PIF3 stabilisation in darkness[11]. A similar interaction has since been observed for PIF5 in dark-grown plants[17]. We hypothesised that UVR8 may promote PIF5 degradation in UV-B by sequestering COP1 and thereby reducing PIF5 stability. We first analysed the abundance of COP1 in our experimental conditions. Consistent with previous reports[18], COP1 abundance was elevated in UV-B in both L*er* and *LerPIF5ox* lines in a UVR8-dependent manner (Fig. 5b, Supplementary Fig. 6b). We next investigated whether COP1

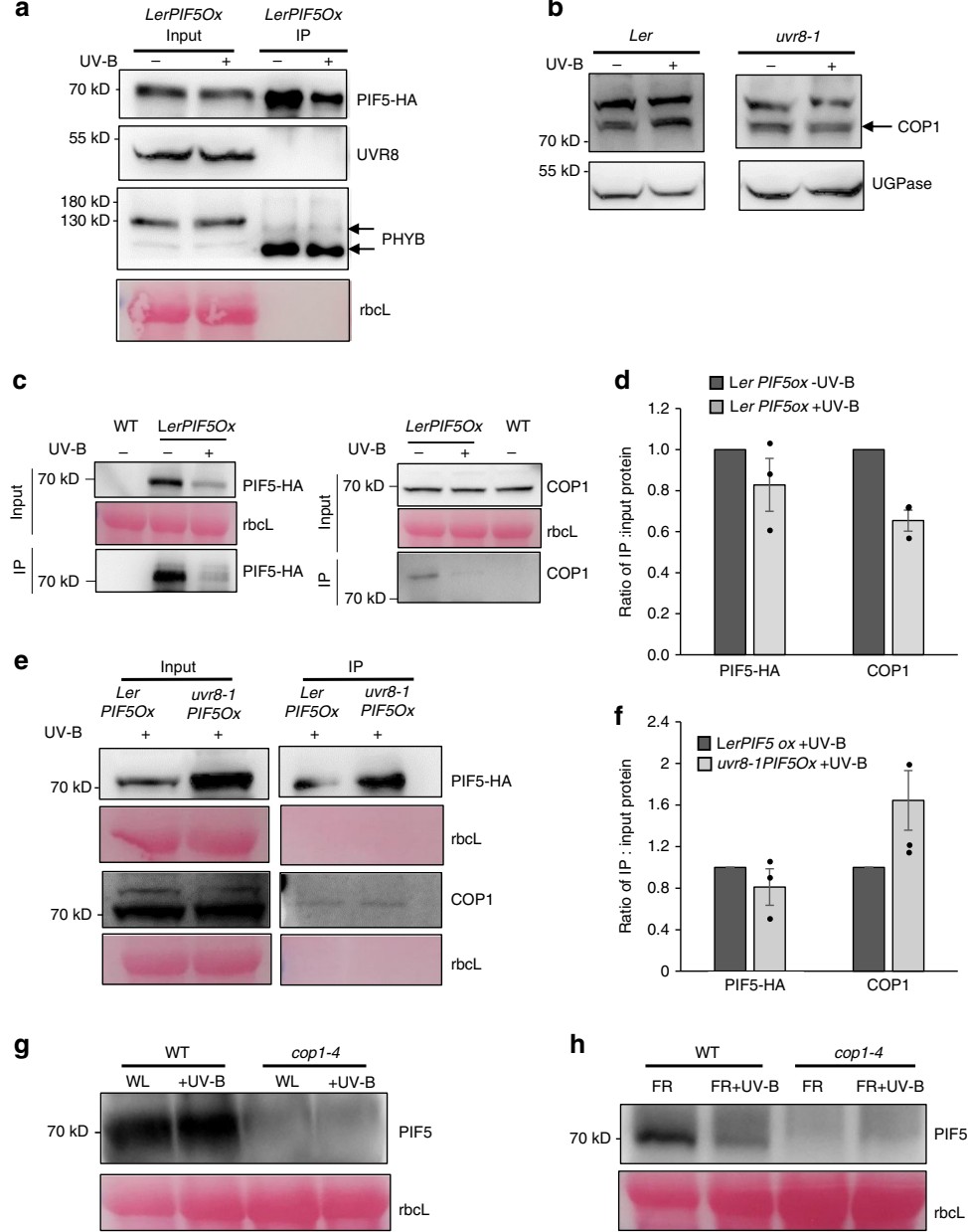

**Fig. 5** PIF5 does not interact with UVR8 but binds to COP1, destabilising PIF5. **a** Co-IP assay showing that PIF5 does not interact with UVR8 *in planta*. *LerPIF5Ox* Seedlings were grown for 10 days in16 h light/8 h dark cycles before transfer at dawn to high R:FR ± UV-B for 1 h. Total protein extracts were immunoprecipitated with anti-HA beads and immunoblots analysed with anti-HA or anti-UVR8 antibodies. An anti-PHYB antibody was used as positive control for PIF5 immunoprecipitation. Ponceau stained Rubisco large subunit (rbcL) was used as a loading control. **b** Western blot of COP1 protein abundance in *Ler* and *uvr8-1*. Seedlings were grown as in (**a**) and immunoblots probed with anti-COP1 and anti-UGPase antibodies. **c** Co-IP assay showing that PIF5 interacts with COP1 in the presence and absence of UV-B. *LerPIF5Ox* and WT seedlings were grown as in (**a**). Immunoblots were probed with anti-COP1 or anti-HA antibodies. Ponceau staining of the Rubisco large subunit (rbcL) was used as a loading control. **d** Quantification of IP/input protein ratio in (**c**) Mean values from three biological repeats are shown. Bars represent SE. **e** Co-IP assay performed as in (**c**) showing increased PIF5-COP1 complex in *uvr8-1* mutants in UV-B. *LerPIF5Ox* and *uvr8-1PIF5Ox* seedlings were grown as in (**a**). **f** Quantification of IP/input protein ratio in (**e**). Mean values from three biological repeats are shown. Bars represent s.e.m. **g**, **h** Western blots of PIF5 protein abundance in 10-day-old Col-0 and *cop1-4* seedlings. Plants were grown in 16 h light/8 h dark cycles before transfer at dawn to high R:FR (WL) ± UV-B (**e**) or low R:FR (FR) ± UV-B (**f**) for 2 h. Immunoblots were probed with an anti-PIF5 antibody. Ponceau staining of the Rubisco large subunit (rbcL) was used as a loading control. Source data are provided as a source data file

interacts with PIF5 in light-grown plants and the impact of UV-B on this process. Co-IP assays were performed with *LerPIF5Ox* seedlings treated with and without UV-B for 1 h (Fig. 5c, d) and UV-B-treated *uvr8-1PIF5Ox* seedlings (Fig. 5e, f). PIF5 degradation is clearly visible at this timepoint (Fig. 2). No PIF5 or COP1 bands were detected in WT controls (Fig. 5c)). Consistent

with previous observations, PIF5 levels were reduced in UV-B-treated *LerPIF5Ox* seedlings and PIF5-HA immunoprecipitations (Fig. 5c). PIF5-COP1 complexes were detected in UV-B-treated and -untreated seedlings, with reduced abundance following UV-B treatment (Fig. 5c, d). In accordance with antibody control tests (Supplementary Fig. 6a), elevated PIF5 was detected in

UV-B-treated *uvr8-1PIF5Ox* seedlings, when compared to *Ler-PIF5Ox* controls (Fig. 5e). A corresponding increase in the proportion of COP1 bound to PIF5 was also observed (Fig. 5f). These data suggest a role for UVR8 in depletion of the PIF5-COP1 complex in UV-B.

The involvement of COP1 in PIF5 stabilisation during shade avoidance was investigated via western blot analysis of native PIF5 abundance in wild-type and *cop1-4* mutants grown in high and low R:FR and treated with supplementary UV-B for 2 h (Fig. 5g, h, Supplementary Fig. 6d, e). Consistent with previous experiments (Fig. 2), PIF5 levels were reduced in wild-type plants following UV-B treatment and this response was exacerbated in low R:FR. PIF5 levels were severely depleted in *cop1* mutants in both high and low R:FR (Fig. 5g, h). PIF5 abundance was so low that further UV-B-mediated reductions in PIF5 abundance could not be detected. Parallel analyses of transcript abundance showed a significant *PIF5* reduction in *cop1* mutants in high, but not low R:FR. This was not further reduced by UV-B treatment (Supplementary Fig. 7). These data suggest that reduced *PIF5* transcript contributes, in part, to the severely reduced abundance of PIF5 observed in *cop1* mutants in high R:FR. The extremely low levels of native PIF5 observed in *cop1* mutants likely contributes to their very short hypocotyls which are not elongated by low R:FR or further inhibited by UV-B (Supplementary Fig. 8).

## Discussion

The role of PIF5 as a key regulator of stem elongation in Arabidopsis is well established. Diurnal growth rhythms of Arabidopsis hypocotyls involve an external co-incidence of high *PIF5* transcript, regulated by the circadian clock, and high PIF5 protein stability, resulting in maximum growth towards the end of the night[19]. Furthermore, the promotion of hypocotyl elongation during shade avoidance has been shown to involve stabilisation of PIF5[13], which, together with PIF4 and PIF7, binds to the promoters of auxin biosynthesis genes, driving auxin biosynthesis[4,5]. PIF5 has also been suggested to increase auxin sensitivity[20]. The ability of plants to enhance their sensitivity to auxin may be important in deep shade, where resources for auxin biosynthesis are limiting and auxin sensitivity increases[21].

Despite the importance of shade avoidance in mixed stands, excessive stem elongation can increase susceptibility to lodging and reduce plant survival[22,23]. Plants have therefore evolved multiple mechanisms to attenuate this response. The activation of phyA in deep shade limits elongation growth, in part, through reducing auxin signalling by direct binding to Aux/IAA proteins[22,24]. PIFs additionally promote the expression of negative regulators of shade avoidance, including LONG HYPOCOTYL IN FAR-RED (HFR1)[25] and PHYTOCHROME RAPIDLY REGULATED 1 (PAR1) and PAR2[26], which form heterodimers with PIF4 and PIF5 and antagonise excessive stem elongation. In addition to these feedback loops in low R:FR signalling, UV-B is a potent inhibitor of growth, providing an unambiguous sunlight signal to supress shade avoidance following sunflecks or emergence from a canopy[6,27]. UV-B-mediated inhibition of shade avoidance has been shown to involve degradation of both PIF4 and PIF5, although the role of UVR8 in this process was not established[6]. Here we show, via the construction of transgenic *PIF5 ox* lines in the *uvr8-1* null background (Fig. 1), that UV-B-mediated PIF5 degradation is rapid and most pronounced in low R:FR, conditions in which PIFs are stabilised (Fig. 2). For the first time, we confirm a role for the UVR8 photoreceptor in this process (Fig. 3).

PIFs 1, 3 and 4 are phosphorylated in the light, leading to ubiquitination by E3 ligase complexes and degradation by the 26S proteasome pathway[28]. PIF3 phosphorylation has been shown to

involve Photoregulatory Protein Kinases 1-4 (PPK1-PPK4)[29], with an additional role for phytochrome Serine/Threonine kinase activity also proposed[30]. Phosphorylation of PIF3/PIF4 and PIF1 have been shown to involve BRASSINOSTEROID-INSENSITIVE 2 (BIN2)[11,31] and Casein Kinase II (CK2), respectively[32]. Ubiquitination of PIFs involves CULLIN (CUL) RING UBIQUITIN LIGASES. Substrate recognition components include EIN3 BINDING F-BOX (EBF1/2) and LIGHT-RESPONSE BRIC-A-BRACK/TRAMTRACK/BROAD (LRB) for PIF3, BLADE ON PETIOLE (BOP1/2) for PIF4 and COP/SPA for PIF1[33–36]. Phytochromes and cryptochromes predominantly control PIF abundance and activity via direct physical interaction[15,37]. In contrast, and in agreement with Y2H studies[6], no physical interaction between UVR8 and PIF5 could be detected *in planta*, despite clear detection of PHYB (Fig.5a). These data suggest that UVR8 may regulate PIF-mediated growth differently to other photoreceptors, although involvement of the 26 S proteasome remains conserved (Fig. 4a–c). In contrast to its established role in protein degradation, the COP/SPA complex has been shown to directly bind to and stabilise PIF3 in the dark[11]. More recently, COP1 has been shown to physically interact with PIF5. This interaction stabilises PIF5 in the dark but promotes its ubiquitination and degradation following transfer to red light[17]. We therefore questioned whether PIF5 bound to COP1 in de-etiolated plants and examined the effect of UV-B on this process. Our immunoprecipitation data showed clear PIF5-COP1 interaction in light-grown seedlings and a reduction of PIF5-COP1 complex in UV-B (Fig. 5c, d). Data showing increased PIF5-COP1 complex in *uvr8-1* mutants in UV-B suggest that activated UVR8 performs a role in depleting PIF5-COP1 complex abundance (Fig. e, f).

The importance of COP1 in stabilising PIF5 in light-grown plants was investigated using western blotting. *cop1* mutants displayed extremely low levels of PIF5 protein in high and low R:FR (Fig. 5g, h). Reduced levels of PIF5 in *cop1* mutants grown in low R:FR are consistent with observations showing reduced PIF1, 3, 4 and 5 levels in *cop1* mutants grown in darkness and suggest a key role for COP1 in stabilising PIFs in conditions with low levels of active phyB[12,17,28]. Reduced levels of PIF5 in *cop1* mutants grown in high R:FR are in agreement with studies of de-etiolated plants[17] but in contrast to a de-etiolation experiment in the same study which showed COP1 to promote PIF5 degradation following transfer from dark to red light[17]. The role of COP1 in controlling PIF5 stability in the light may therefore differ between de-etiolating seedlings and fully de-etiolated plants. The requirement for the N-terminus of PIF5 for UV-B-mediated degradation supports data showing that both N- and C-terminal regions of PIF3 interact with SPA1 and therefore the COP/SPA complex[11] (Fig. 4d–g). It is possible that, in high R:FR, phyB competes with COP1 for PIF5 binding sites. Sequestration of COP1 by UVR8 would then facilitate established phyB-mediated PIF5 degradation by phosphorylation and ubiquitination[13]. The stabilising effect of COP1 on PIF5 in low R:FR, where reduced levels of active phyB exist does, however, suggest that COP1 binding may stabilise PIF5 by other mechanisms in addition to outcompeting phyB binding. One possibility is that COP1 promotes the activity of the TOPP4 protein, involved in PIF5 dephosphorylation[38].

The severely impaired shade avoidance response of *cop1* mutants is likely explained by reduced PIF levels, in addition to accumulation of the PIF inhibitor, HFR1 (Supplementary Fig. 8)[39,40]. Collectively, our data support a model whereby COP1 stabilises PIF5 (and possibly other PIFs) in low R:FR to drive shade avoidance. Indeed, COP1 has been shown to re-accumulate in the nucleus in these conditions[16]. Although not the focus of this study, SPA proteins have been shown to affect the

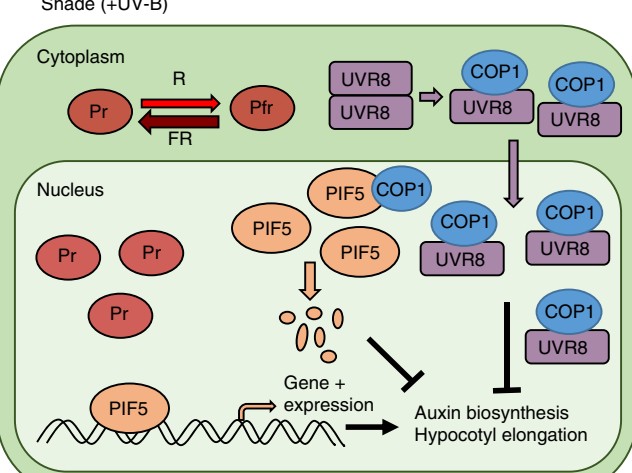

**Fig. 6** Hypothetical model depicting how UVR8 regulates PIF5 abundance and hypocotyl elongation in low R:FR. When shaded, the photoequilibrium of phytochrome shifts towards the biologically inactive Pr form. This results in the stabilisation of PIF5 which is further enhanced by binding to COP1. PIF5 promotes the expression of genes involved in auxin biosynthesis and hypocotyl elongation. When sunlight is reached, UVR8 dimers absorb UV-B, causing them to monomerise and bind COP1. UVR8-COP1 complexes enter the nucleus and promote UV-B signalling which inhibits auxin biosynthesis and hypocotyl elongation via multiple mechanisms (Hayes et al. 2014). In addition, the sequestration of COP1 by UVR8 destabilises PIF5, further suppressing hypocotyl elongation

light responsiveness of COP1 sub-cellular localisation in addition to their role in promoting COP1 activity[41]. Upon detection of sunflecks or emergence through the canopy, UVR8-mediated detection of UV-B promotes rapid degradation of PIF5 to limit shade avoidance and prevent unnecessary stem elongation once sunlight has been reached (summarised in Fig. 6). UVR8-mediated inhibition of PIF signalling via DELLA stabilisation[6] acts over a longer time frame to promote resource reallocation towards leaf development and photosynthesis in sunlight. The role of COP1 and SPA proteins in regulating PIF4 and PIF7 abundance/activity in low R:FR remains a key question for future research.

## Methods

**Plant materials and growth conditions**. Landsberg *erecta* (L*er*) and Columbia-0 (Col) accessions of *Arabidopsis* were used as wild-type controls in this study.

Transgenic L*er* and *uvr8-1* lines expressing *35S:PIF5-HA* were constructed using *Agrobacterium tumefaciens* floral-dip. The *35S:PIF5-HA* construct pCF404 was provided by Professor Christian Fankhauser[13]. Transgenic seeds were screened for fluorescence under a Leica MZFLIII microscope using a GFP2 filter (460–500nm-510 LP). Transformants displaying a 3:1 segregation ratio were self-fertilised and homozygous progeny tested for PIF5-HA protein expression. *uvr8-6*, *cop1-4*, *pif4-101*, *pif5*, *pif7-1*, *pif4-101/pif5*, *pif4-1/pif7-1* and *pifq* are in the Col-0 background[42–45]. *uvr8-1* and *phyB-1* are in the L*er* background[46,47]. *35S:ΔNPIF5-HA* lines and corresponding *35S:PIF5-HA* lines are in the Col-background and lack the first 68 amino acids of the PIF5 protein[13].

*Arabidopsis* seeds were sown directly onto a 3:1 mixture of compost and horticultural silver sand. After 3 days stratification in darkness at 4 °C, seeds were germinated in controlled growth cabinets (Microclima 1600E, Snijder Scientific, The Netherlands) in continuous white light (R:FR ~ 8.0) or under 16 h light /8 h dark cycles at 20 °C and 70% humidity. White light was provided by cool-white fluorescent tubes (400–700 nm) at photon irradiance of 80 µmol m$^{-2}$ s$^{-1}$. Supplementary narrowband UV-B (~1.0 µmol m$^{-2}$ s$^{-1}$) was provided by Philips TL100W/01 tubes. Supplementary FR LEDs positioned overhead (peak emission 735 nm) reduced R:FR to 0.06 for low R:FR experiments. All light measurements were performed using an Ocean Optics FLAME-S-UV–VIS spectrometer with a cosine corrector (oceanoptics.com).

**Hypocotyl measurements**. Seedlings were grown in continuous WL for 3 days then moved to either high or low R:FR ± UV-B for 4 days. Hypocotyls were measured in Image J (http://rsb.info.nih.gov/ij/).

**Determination of *PIF5* transcript levels**. Seedlings were grown in 16 h light/8 h dark cycles for 10 days, before transfer at dawn to different light conditions for the indicated time. Approximately 50 µg of aerial tissue was harvested into liquid nitrogen at predawn and at indicated times in light conditions. RNA was extracted using a spectrum total RNA kit (Sigma) according to the manufacturer's instructions. This was reverse transcribed using a High Capacity cDNA Reverse Transcription kit (Applied Biosystems). Real-time PCR reactions were performed with 2X Brilliant III SYBR Green QPCR (Agilent Technologies) and data analysed using MxPro software (Agilent Technologies). Transcript levels were normalised to *ACTIN2*. Primer sequences are provided in Supplementary Table 1.

**Protein extraction and immunoblots**. Frozen samples were ground into fine powder then mixed with extraction buffer (50 mM Tris-HCl, pH 7.5, 150 mM NaCl, 1% Na deoxycholate, 0.5% (v/v) Triton X-100, 1 mM DTT, 10 µl/ml Sigma protease inhibitor cocktail, 50 µM MG132). After centrifugation at 14,000*g* for 10 min at 4 °C, proteins in supernatants were quantified using a Bradford assay (Bio-Rad). 40 µg of protein was mixed with SDS-PAGE sample buffer (4 × 250 mM Tris HCl pH 6.8, 2% SDS, 40% (v/v) glycerol, 20% (v/v) β-mercaptoethanol, 0.5% bromophenol blue), and heated for 5 min at 95 °C before resolving on 10% SDS-PAGE gels. Proteins were transferred to PVDF membrane and visualised by staining with Ponceau S. The membrane was cut across the 55 kDa region and blocked in 10% skimmed milk powder in TBS-T for 2 h. For PIF5-HA detection, the upper membrane was incubated in a 1: 2500 dilution of anti-HA antibody conjugated to peroxidase (Roche 12013819001) and the bottom membrane in a 1:5000 of anti-UGPase antibody (Agrisera) overnight at 4°C. UGPase blots were further incubated in a 1: 30,000 dilution of anti-rabbit antibody (Promega). Signals were detected using SuperSignal West Femto maximum sensitivity substrate (Thermo Fisher) and visualised using a Fusion Pulse imager (Vilber Lourmat). For COP1 detection, a 1:500 dilution of anti-COP1 antibody was used[48], followed by a 1:5000 dilution of anti-rabbit antibody. For native PIF5 detection, polyclonal antibody was produced in rabbit by GenScript using the full length PIF5 sequence (AT3G59060) with an N-terminal 6xHis tag. The specificity of affinity-purified anti-PIF5 antibody was confirmed via western blotting with *pif5* mutant and *PIF5ox* lines (Supplementary Fig. 6c, d). For native PIF5 immunoblots, a modified extraction and blotting procedure was followed. Frozen tissue samples were ground into fine power then mixed with extraction buffer (100 mM MOPS (pH 7.6), 40 mM β-mercaptoethanol, 5% SDS, 4 mM EDTA, 2 mM PMSF, 10 µl/ml protease inhibitor cocktail) and boiled for 3 min at 90°C. After centrifugation at 14,000*g* for 10 min, the protein concentrations of supernatants were quantified. 50 µg protein was mixed with SDS-PAGE sample buffer and heated for 5 min at 95 °C before resolving on 8% SDS-PAGE gels. Blots were blocked in SEA BLOCK blocking buffer (Thermo Fisher) for 2 h before incubation in anti-PIF5 antibody (1:2000) overnight at 4°C. Membranes were then incubated in anti-rabbit-HRP antibody (1:10,000) for 1 h at room temperature. For protein quantification, EvolutionCapt software was used to determine the density of bands on immunoblots, using exposure times with unsaturated signals.

**Proteasome inhibition**. Plants were grown in 16 h light/8 h dark cycles of WL for 10 days then transferred to one-half strength MS liquid medium containing MG132 (50 µM dissolved in 0.1% (v/v) dimethyl sulfoxide) and incubated for 16 h. At dawn, plants were transferred to low R:FR or low R:FR + UV-B for 40 min. Control plants were transferred to ½ strength MS containing 0.1% DMSO. Protein extraction and immunoblots were performed as described above. An anti-ubiquitin

antibody (Abcam, ab7254) was used at a 1:2000 dilution as a positive control to confirm that MG132 had imbibed in to plant tissues and confirm protein ubi-quitination in PIF5 immunoprecipitates. Blots were subsequently incubated in a 1:10000 dilution of anti-mouse antibody (Dako).

**Co-immunoprecipitation.** Plants were grown in 16 h light/8 h dark cycles of WL for 10 days then transferred to different light conditions for 1 h. Ten grams of aerial tissue was harvested in liquid nitrogen and homogenised in 4 ml extraction buffer (50 mM Tris-HCl, pH 8, 150 mM NaCl, 0.5% (v/v) Nonidet P-40, 0.05% sodium deoxycholate, 10 mM DTT, 1 mM PMSF, 1 mM EDTA, 1.5x protease inhibitors (Sigma)). Extracts were centrifuged twice at 14,000g for 15 min at 4 °C to remove cell debris. Total protein was quantified with a Bradford assay (Bio-Rad) and 4 mg protein incubated with 50 μl of anti-HA magnetic beads (μMACS Epitope Tag, Miltenyi Biotec) for 3 h in cold room with gentle rotation. A small aliquot of protein sample was kept aside for loading as input controls. Protein samples with beads were loaded into a 20μMACS® Separation Column (Miltenyi Biotec) equi-librated with 200 μl of extraction buffer and placed in the μMACS separator (Miltenyi Biotec). The unbound fraction was collected, and columns washed 4 times with 200 μl extraction buffer. 20 μl elution buffer was heated to 95 °C and added to the column for 5 min. Bound proteins were then eluted in 85 μl heated elution buffer. For immunoblot analysis, 60 μg of protein was loaded for both input and eluted fractions (IP) on 10% SDS-PAGE gels and transferred to PVDF membranes. Membranes were blocked in 10% skimmed milk powder in TBST for 2 h and probed with antibodies overnight at 4 °C. Incubation in anti-HA antibody was used to confirm PIF5 immunoprecipitation. Blots were probed with an anti-PHYB antibody as a positive control. This consisted of a 1:40 dilution of B1 and B7[49] followed by incubation in a 1:2000 dilution of anti-mouse antibody. UVR8 was detected using a 1:10000 dilution of a polyclonal UVR8 antibody[50] followed by incubation in a 1:20000 dilution of an anti-rabbit antibody. For co-immunoprecipitation experiments, a higher concentration of COP1 antibody was used than for western blots (1:200). Chemiluminescence signals were detected as described above. For immunoprecipitated PIF5-HA and COP1 quantification, UV-B-untreated (Fig. 5d) and LerPIF5ox (Fig. 5f) samples were selected as references and given a value of 1. Relative signal values of UV-B-treated (Fig. 5d) and uvr8PIF5ox (Fig. 5f) samples were then determined. Input values were normalised to ponceau-stained rbcL to account for slight variations in loading before IP/input calculations were performed.

**Statistical analyses.** Statistical analyses were performed using IBM SPSS Statistics 24.0 software. All hypocotyl length experiments were repeated three times and one representative data set displayed. Hypocotyl measurements were analysed using a one-way ANOVA and Tukey's HSD test. For transcript analyses, relative abun-dance values were first transformed by log-2. Student's t-tests were performed to investigate significant difference between the means indicated in the figure legends ($p < 0.05$).

**Reporting summary.** Further information on research design is available in the Nature Research Reporting Summary linked to this article.

## Data availability

The datasets generated and analysed in the current study are provided as a Source Data file. Other supporting data are available from the corresponding author upon request. There are no restrictions on data availability.

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

## Acknowledgements

The authors thank Professor Christian Fankhauser (Lausanne) for the donation of mutants (*pif4*, *pif5*, *pif4/5*), transgenic lines (*35S:PIF5-HA, 35S:ΔNPIF5*-HA) and the pCF404 construct. We also thank Professor Peter Quail (PGEC) for the donation of mutants (*pif7-1*, *pif4/7* and *pifq*). We thank Kester Cragg-Barber for technical assistance. This work was funded by BBSRC grants BB/M008711/1 and BB/R002045/1 to KAF and GIJ and grant HO2793/3-2 from the Deutsche Forschungsgemeinschaft to U.H. SH was supported by a NERC studentship.

## Author contributions

AS, SH, KK, UH, GIJ and KAF designed experiments. AS, BS, SH and KK performed experiments. All authors analysed data and wrote the manuscript.

## Competing interests

The authors declare no competing interests.
