## [Peer Review File · Nature Communications]

Reviewers' comments:

Reviewer #1 (Remarks to the Author):

The manuscript by Sharma et al. demonstrates that UV-B inhibits shade avoidance by inducing PIF5 destabilisation via the ubiquitin-proteasome system. The UV-B photoreceptor UVR8 mediates the rapid degradation of PIF5 via sequestration of COP1 which promotes PIF5 stabilisation in low R:FR. This study is potentially interesting. However, the evidence is not quite consistent with their previous study, and is not convincing based on limited biochemical assays with less suitable data quality. The study needs substantial work to obtain clear and convincing results for the molecular mechanism before it can be considered for publication.

1. It has been reported that UV-B-mediated inhibition of shade-induced hypocotyl elongation involves PIF4 and PIF5 (Hayes et al., 2014, PNAS). The authors consistently found in this study that PIF5 overexpression led to long hypocotyl under WL and WL+UV-B compared with WT (Fig. 1a). However, similarly with WT, LerPIF5Ox showed the inhibition of hypocotyl elongation in response to UV-B. Therefore it is not clear about the relative contribution of PIF5 in this process as well as the significance of this study.

2. The inhibition of hypocotyl elongation (the ratio of WL+UV-B versus WL) looks similar or a little bit lower in LerPIF5Ox than in Ler (Fig. 1a). However, the ratio is much higher in 35S:PIF4-HA and 35S:PIF5-HA than Col (Fig. S7C and S7D in Hayes et al., 2014, PNAS). The regulation of PIF4/PIF5 is ecotype or light condition dependent? But the phenotype of *pif4/pif5* (Fig. S7A in Hayes et al., 2014, PNAS) is consistent with the model proposed by this study. The reviewer is confused.

3. As claimed by the authors, the molecular mechanism of PIF5 destabilisation is basically relied on the interaction between UVR8 and COP1 which removes the stabilisation of PIF5 by COP1. However, there is no evidence showing the specific disruption of the UVR8-COP1 interaction makes PIF5 stable. Data from *uvr8* and *cop1* mutants did not provide convincing information on this mechanism, as they affect almost all the UV-B responses at a genome-wide level.

4. The authors examined the interaction between COP1 and PIF5 and investigated the impact of UV-B on this process (Fig. 5c). However, no information about the impact of UV-B was provided as there was no control of -UV-B like Fig. 5a. More, COP1 protein level was higher in *uvr8-1* PIF5Ox than LerPIF5Ox in the input fraction, thus more COP1 in the IP fraction of *uvr8-1* PIF5Ox cannot support the conclusion that UVR8-mediated sequestration of COP1 destabilises PIF5 (Fig. 5c seems the only result meant to support this mechanism).

5. Reduced PIF5 abundance was found in *cop1-4*, but it is not clear whether this is resulted from low stability (should be restored with MG132 treatment) or low transcripts. In *cop1-4*, PIF5 protein level is already low (almost undetectable) under FR, and looks equivalent under FR and FR+UV-B (Fig. 5d). That is, UV-B did not further destabilise PIF5? The stabilisation of PIF5 by COP1 is independent of UV-B? This result seems contradictory to this study.

6. From the immunoblots in Fig. 3a, the degradation is PIF5-HA in *uvr8-1* PIF5Ox is very obvious at various time points under both WL and WL+UV-B conditions. Why the conclusion is "supplementary UV-B had no effect PIF5 abundance, confirming the role of UVR8 in mediating this response". It seems that the degradation occurred independent of UV-B and UVR8.

7. To analyze whether PIF5 degradation occurs via the proteasome system, the authors used an anti-ubiquitin antibody to blot total proteins of various materials (Fig. 4a), which did not present the ubiquitination state of PIF5 protein. MG132 treatment usually results in an overall increase in ubiquitination. To do this, the immunoprecipitation should be performed to specifically target PIF5 by using anti-HA and then use anti-ubiquitin for western blot.

Reviewer #2 (Remarks to the Author):

Comments:

This study reveals that UV-B promotes the degradation of PIF5 dependent on UVR8. COP1 interacts with and stabilizes PIF5 in low R:FR, while UV-B promotes UVR8-COP1 interaction to destabilize PIF5 whose stability is dependent its APB domain. This study aims to answer an important question in the field. It would be a suitable work for NC if following issues be addressed, to substantiate the conclusions.

1. In the previous study by Hayes et al. (Hayes et al., PNAS, 2014), *pif4/pif5* mutant showed shorter hypocotyl length under WL+UV-B+FR than WL+FR. This result indicated UV-B-mediated inhibition of low R:FR-induced hypocotyl elongation was not entirely dependent on PIF4 and PIF5. Thus, it's necessary to clarify whether UV-B-mediated inhibition of high or low R:FR-induced hypocotyl elongation in Ler (Fig.1a and b) was dependent on PIF5 protein level. It would be more satisfactory if the endogenous PIF5 protein levels in Ler under different light conditions be examined.
2. The previous study used PIF5-overexpressing lines in the Col-0 background (Hayes et al., 2014, PNAS). It is interest to note the authors reconstruct 35S:PIF5-HA in Ler background in this study? Are the phenotypes and biochemical results derived from 35S:PIF5-HA in Ler comparable to those in Col-0?
3. The authors concluded that UVR8-mediated sequestration of COP1 in UV-B reduces the stability of PIF5, providing a mechanism for rapid reduction of PIF5 abundance in sunlight. More analysis, such as the effect of loss-of-function *cop1* mutation (e.g. *cop1-4*) should be examined in high R:FR (WL) and low R:FR (FR) conditions supplemented with or without UV-B. Further more, the phenotype of PIF5-overexpressing lines in *cop1* in high R:FR (WL) and low R:FR (FR) conditions supplemented with or without UV-B could be examined as well. A recent study that the COP1-SPA complex is necessary for the red light-induced degradation of PIF5 by using PIF5-Myc and PIF5-Myc/*cop1-4* has been reported (Pham et al., Plant Journal, 2018) and can be a good example.
4. COP1 has been shown to physically interact with PIF5 and promotes the ubiquitination and degradation of PIF5 in red light (Pham et al., Plant Journal, 2018). In this study, Fig. 5c shows that COP1 associates with PIF5 in high R:FR. Does COP1 also promote PIF5 degradation in high R:FR? If so, based on the model proposed in this study, UV-B induced UVR8 sequestration of COP1 will inhibit COP1 mediated PIF5 degradation in high R:FR+UV-B. That is, PIF5 will be stabilized in high R:FR+UV-B. However, Fig. 2 shows PIF5 is rapidly degraded in high R:FR+UV-B. This point require close attention.
5. For phenotypic analysis in Fig.1, the seedlings were grown in high R:FR for 3 days before UV-B treatment. But for biochemical analysis in Figs. 2-5, the seedlings were grown in 16 h light/8h dark cycles for 10 days before UV-B treatment. Why different conditions were used?
6. "Immunoprecipitated PHYB was a lower molecular weight, suggesting that phyB-PIF5 interaction results in modification of the PIF5 protein (Fig. 5a)." PHYB of lower molecular weight led to PIF5 modification? It is not clear how the conclusion is made. Please clarify it. Since PHYB constitutively interacts with PIF5 (Fig. 5a), does PHYB-PIF5 interaction interfere COP1-PIF5 interaction and PIF5 stability? Though PIF5 does not interact with UVR8 as reported (Hayes et al., PNAS, 2014), does UVR8 affect the physical interaction between COP1 and PIF5.
7. PHYB has been shown to interact with the APB domain of PIF5 in vitro and promote its degradation in red light. In this study, the authors reveal that PHYB constitutively interacts with PIF5, and that the N-terminal ABP domain of PIF5 is required for UV-B-mediated degradation. However, it has been reported that UV-B-mediated inhibition of low R:FR-induced hypocotyl elongation occurs independent of phytochrome (Hayes et al., PNAS, 2014). Are those results consistent with each other?
8. The immunoblots in Figs. 2-5 did not include corresponding WT or mutant lines as negative control. In Fig. 4b, the immunoblot using ubiquitin blot reflected the ubiquitination of total proteins, but did not reflect the ubiquitination level of PIF5-HA.

Reviewer #3 (Remarks to the Author):

This ms reported a study of the molecular mechanism controlling PIF5 degradation in response to UV-B under low R/FR condition. The authors showed that UV-B induced PIF5 protein degradation was mediated by UVR8 photoreceptor through ubiquitin-proteasome pathway, and the N-terminus phytochrome-interacting domain of PIF5 was needed for its degradation. They further proved that PIF5 interacted with COP1 in low R/FR shade-grown plants treated with UV-B to inhibit PIF5 degradation. Given that COP1 is a known partner of UV-B-photoexcited UVR8 monomer, the authors argued that UVR8 mediates PIF5 degradation in response to UV-B light, by sequestration of the PIF5 stabilizer: COP1. This seems an interesting hypothesis because it seems distinct with the previously reported stabilizing effects of COP1 on other PIF proteins. The manuscript is well written that addresses a question of general interest in plant photobiology. The data presented are based on some well-designed experiments which pertinently support the conclusion.

Specific comments

1. It is not clear whether UV-B affects COP1-PIF5 interaction. The authors may want to address this issue by a Fig. 5c-related experiment in the presence or absence of UV-B light.
2. Line 171, "Immunoprecipitated PHYB was a lower molecular weight, suggesting that phyB-PIF5 interaction results in modification of the PIF5 protein (Fig. 5a)". This sentence does not seem to make much sense. Because the blot shown in Fig. 5a seem to indicate that PIF5 pulled down both slow and fast migrating phyB bands, it may be an artifact of phyB degradation due to the prolonged IP reaction. Regardless, why does this observation argue for "phyB-PIF5 interaction results in modification of the PIF5 protein"? Third, there seems no different effects of UV-B, then how is this experiment associated with the main thesis of this ms?
3. The authors seem to use Fig. 4b to show the effects of MG132. It may be interesting to show ubiquitylation of the IP products of PIF5.

Reviewer #4 (Remarks to the Author):

This manuscript uncovers a novel mechanism by which plants integrate UVB light signals into the complex network they use to adapt their growth and development to a constantly fluctuating light environment. It is becoming increasingly clear that multiple signals converge in plant cells through the small PIF transcription factor family, to regulate a target transcriptional network that transduces these signals into adaptational phenotypic responses. This work from Sharma et al. adds an important new dimension to that picture by showing how the UVB receptor, UVR8, exploits a non-canonical function of an E3 ubiquitin ligase, COP1, to rapidly regulate the abundance of the PIF5 transcription factor in response to UVB light levels.

My only comment is, that it seems to me that COP1 could be acting antagonistically (in dynamic fashion) to the established phyPfr-induced phosphorylation-ubiquitination-degradation mechanism, by interfering with the binding of phy, kinases and/or ligases to PIF5. The UVB/UVR8-COP1 sequestration would then represent a rapid mechanism of modulating the rate of the existing phy/kinase/ligase-driven degradation (and therefore steady-state levels) of PIF5 across the fluctuating light/dark/shade/sunfleck/etc light environment. I would like to see some discussion of how these UVB signals might be integrated into this known framework of PIF-abundance regulation.

Some editorial points:

1. Fig. 1b, right panel: There seems to be an extra 'dot' there.
2. Fig. 4 legend: Need to define 'FR' here.
3. Fig. 5d: Need to indicate that these panels are replicate experiments (if that's what they are).
4. Lines 142-4: Should read something like: "Blots were also probed with an anti Ubiquitin antibody

to confirm that the MG132 had fully infiltrated into the tissue and enhanced accumulation of ubiquitinated proteins (Fig. 4b)".

5. Line 157: show/suggest? Need to choose.

6. Line 172: Apparent typo. Should be "suggests modification of the phyB protein".

Reviewer #1 (Remarks to the Author):

The manuscript by Sharma et al. demonstrates that UV-B inhibits shade avoidance by inducing PIF5 destabilisation via the ubiquitin-proteasome system. The UV-B photoreceptor UVR8 mediates the rapid degradation of PIF5 via sequestration of COP1 which promotes PIF5 stabilisation in low R:FR. This study is potentially interesting. However, the evidence is not quite consistent with their previous study, and is not convincing based on limited biochemical assays with less suitable data quality. The study needs substantial work to obtain clear and convincing results for the molecular mechanism before it can be considered for publication.

1. It has been reported that UV-B-mediated inhibition of shade-induced hypocotyl elongation involves PIF4 and PIF5 (Hayes et al., 2014, PNAS). The authors consistently found in this study that PIF5 overexpression led to long hypocotyl under WL and WL+UV-B compared with WT (Fig. 1a). However, similarly with WT, LerPIF5Ox showed the inhibition of hypocotyl elongation in response to UV-B. Therefore it is not clear about the relative contribution of PIF5 in this process as well as the significance of this study.

The reviewer is correct that the elongated hypocotyl phenotype of *LerPIF5ox* lines is suppressed by UV-B, providing clear evidence that UV-B can suppress PIF5 activity. Shade avoidance is predominantly regulated by the combined actions of PIF4, PIF5 and PIF7. Although this manuscript focuses on the regulation of PIF5 abundance we have previously shown that UV-B can also suppress PIF4 abundance (Hayes et al. 2014 PNAS). It is likely that UV-B-mediated inhibition of shade avoidance involves the combined suppression of PIF4, PIF5 and PIF7 abundance and activities. The significance of this study (highlighted by the other reviewers) is that we provide a novel molecular mechanism through which UV-B affects PIF protein abundance. Although beyond the scope of this study, UV-B may be affecting PIF4 and PIF7 abundance similarly. To highlight the effects of UV-B on multiple PIF activities, we have included Figure S1 which shows UV-B-mediated shade avoidance inhibition in a variety of *pif* single and higher order mutants. These data are entirely consistent with published literature and show that PIFs 4, 5 and 7 all regulate hypocotyl elongation in high R:FR. They additionally confirm that PIF7 is the predominant PIF regulating shade avoidance, with additional roles for PIF4 and PIF5. UV-B-mediated suppression of hypocotyl elongation in *pif4/7* mutants in both high and low R:FR supports a role for UV-B-mediated PIF5 turnover in this response. Text has been modified to incorporate these observations.

2. The inhibition of hypocotyl elongation (the ratio of WL+UV-B versus WL) looks similar or a little bit lower in LerPIF5Ox than in Ler (Fig. 1a). However, the ratio is much higher in 35S:PIF4-HA and 35S:PIF5-HA than Col (Fig. S7C and S7D in Hayes et al., 2014, PNAS). The regulation of PIF4/PIF5 is ecotype or light condition dependent? But the phenotype of *pif4/pif5* (Fig. S7A in Hayes et al., 2014, PNAS) is consistent with the model proposed by this study. The reviewer is confused.

We are not convinced that the direct comparison of data values (the ratio of hypocotyl length between two light treatments) is meaningful between different studies with slightly different experimental conditions (PAR is 90 $\mu\text{molm}^{-2}\text{s}^{-1}$ in Hayes et al and 80 $\mu\text{molm}^{-2}\text{s}^{-1}$ here) using different transgenic lines (Col and Ler) with different levels of *PIF5* expression. We strongly

disagree that this value should be compared between *35S:PIF4-HA* lines in Col (Hayes et al. 2014) and *35S:PIF5-HA* lines in *Ler* (this study) as this presents 4 differing variables (the identity of the overexpressed gene, the level of overexpression, the background accession and the PAR value used). What is clear is that all lines examined (*35S:PIF4-HA* in Col, *35S:PIF5-HA* in Col and *35S:PIF5-HA* in *Ler*) display an elongated hypocotyl phenotype which is suppressed by UV-B.

3. As claimed by the authors, the molecular mechanism of PIF5 destabilisation is basically relied on the interaction between UVR8 and COP1 which removes the stabilisation of PIF5 by COP1. However, there is no evidence showing the specific disruption of the UVR8-COP1 interaction makes PIF5 stable. Data from *uvr8* and *cop1* mutants did not provide convincing information on this mechanism, as they affect almost all the UV-B responses at a genome-wide level.

We are unsure how to address this query. UVR8-COP1 interaction is central to UVR8 signalling, so it is impossible to study disruption of this process, without affecting a significant number of UV-B responses at the genome-wide level.

4. The authors examined the interaction between COP1 and PIF5 and investigated the impact of UV-B on this process (Fig. 5c). However, no information about the impact of UV-B was provided as there was no control of -UV-B like Fig. 5a. More, COP1 protein level was higher in *uvr8-1 PIF5Ox* than *LerPIF5Ox* in the input fraction, thus more COP1 in the IP fraction of *uvr8-1 PIF5Ox* cannot support the conclusion that UVR8-mediated sequestration of COP1 destabilises PIF5 (Fig. 5c seems the only result meant to support this mechanism).

We agree with the reviewer that data showing the effect of UV-B on PIF5-COP1 interaction would be helpful. We have therefore included an additional component within Figure 5, showing COP1 abundance in PIF5 immunoprecipitations extracted from plants treated with and without UV-B. These blots confirm that UV-B reduces the abundance of PIF5-COP1 complex. These data are in accordance with our hypothesised model which suggests that COP1 binds to and stabilises PIF5 in the absence of UV-B. Following UV-B treatment, UVR8 sequesters COP1, resulting in the degradation of 'unbound' PIF5. Text has been modified accordingly.

5. Reduced PIF5 abundance was found in *cop1-4*, but it is not clear whether this is resulted from low stability (should be restored with MG132 treatment) or low transcripts. In *cop1-4*, PIF5 protein level is already low (almost undetectable) under FR, and looks equivalent under FR and FR+UV-B (Fig. 5d). That is, UV-B did not further destabilise PIF5? The stabilisation of PIF5 by COP1 is independent of UV-B? This result seems contradictory to this study.

The reviewer raises an excellent point and we thank them for this helpful suggestion. We have now analysed *PIF5* transcript abundance in WT and *cop1* mutants in high and low R:FR ± UV-B. This is now Shown in Figure S7. In high R:FR, we observed a significant UV-B-mediated reduction in *PIF5* transcript abundance in WT plants. These data are consistent with Supplementary Figure 3 which also shows that the response is mediated by UVR8. A reduction in *PIF5* transcript of approximately a 50% was observed in *cop1* mutants, which was not further reduced following UV-B treatment. In low R:FR, there was no significant difference between WT and *cop1* mutants at 2 h, but some reduction was observed following UV-B treatment. These data suggest that the severe depletion of PIF5 protein we observe in *cop1* mutants in high R:FR results from a combination of reduced *PIF5* transcript abundance

and enhanced protein degradation. The text has been modified to incorporate these new data.

UV-B-mediated degradation of PIF5 could not be observed in *cop1* mutant plants because PIF5 levels are largely undetectable before UV-B treatment (ie. there is no detectable COP1 to degrade). This result is entirely consistent with our model. In the absence of UV-B, COP1 binds to and stabilises PIF5. In UV-B, UVR8 sequesters some COP1, reducing PIF5 stability. We have adjusted the text and the summary model in Figure 6 to make this message clearer.

6. From the immunoblots in Fig. 3a, the degradation is PIF5-HA in *uvr8-1* PIF5Ox is very obvious at various time points under both WL and WL+UV-B conditions. Why the conclusion is “supplementary UV-B had no effect PIF5 abundance, confirming the role of UVR8 in mediating this response”. It seems that the degradation occurred independent of UV-B and UVR8.

PIF5 degradation occurs via phytochrome interaction following dark to WL transfer. This is entirely consistent with published literature and explains the accumulation of PIF5 in low R:FR, where active phytochrome is depleted (eg. Lorrain et al 2008 Plant Journal). UV-B, perceived by UVR8 acts to enhance the rate of PIF5 degradation, possibly in part, due to competition between COP1 and phyB for the N terminal region of PIF5 (an interesting suggestion provided by reviewers 2 and 4). We have therefore modified the text to make these points clearer.

7. To analyze whether PIF5 degradation occurs via the proteasome system, the authors used an anti-ubiquitin antibody to blot total proteins of various materials (Fig. 4a), which did not present the ubiquitination state of PIF5 protein. MG132 treatment usually results in an overall increase in ubiquitination. To do this, the immunoprecipitation should be performed to specifically target PIF5 by using anti-HA and then use anti-ubiquitin for western blot.

We agree with the reviewer that western analysis of PIF5-immunoprecipitates with a ubiquitin antibody would be useful. This experiment was technically challenging but we have included our results in Figure 4c and modified the text accordingly. Western analysis of immunoprecipitated PIF5 showed a ladder of ubiquitinated proteins which was enhanced in the presence of proteasome inhibitor. This was also requested by reviewers 2 and 3.

Reviewer #2 (Remarks to the Author):

Comments:

This study reveals that UV-B promotes the degradation of PIF5 dependent on UVR8. COP1 interacts with and stabilizes PIF5 in low R:FR, while UV-B promotes UVR8-COP1 interaction to destabilize PIF5 whose stability is dependent its APB domain. This study aims to answer an important question in the field. It would be a suitable work for NC if following issues be addressed, to substantiate the conclusions.

1. In the previous study by Hayes et al. (Hayes et al., PNAS, 2014), *pif4/pif5* mutant showed shorter hypocotyl length under WL+UV-B+FR than WL+FR. This result indicated UV-B-mediated inhibition of low R:FR-induced hypocotyl elongation was not entirely dependent on PIF4 and PIF5. Thus, it's necessary to clarify whether UV-B-mediated inhibition of high or low R:FR-induced hypocotyl elongation in Ler (Fig.1a and b) was dependent on PIF5 protein level. It would be more satisfactory if the endogenous PIF5 protein levels in Ler under different light conditions be examined.

UV-B-mediated suppression of shade avoidance is likely to involve suppression of PIF4, PIF5 and PIF7 abundance/activity. Please see response 1 to reviewer 1.

2. The previous study used PIF5-overexpressing lines in the Col-0 background (Hayes et al., 2014, PNAS). It is interest to note the authors reconstruct 35S:PIF5-HA in Ler background in this study? Are the phenotypes and biochemical results derived from 35S:PIF5-HA in Ler comparable to those in Col-0?

Ler was used in this study to incorporate the extensively characterised *uvr8-1* mutant. Significant UV-B-mediated suppression of hypocotyl elongation in *PIF5ox* is observed in both Ler and Col backgrounds, although absolute values cannot be compared due to differences in *PIF5* expression between different transgenic lines and small differences in experimental conditions between studies (Hayes et al. 2014 and Figures 1a, S2 and S3 in this study). Clear UV-B-mediated PIF5-HA degradation can be observed at 2 h in high and low R:FR in both Ler (Figure 2- this study) and Col (Figure 4d- this study, Hayes et al. 2014). We feel that observations showing tagged PIF5 to behave similarly in two Arabidopsis accessions increases the robustness of our findings.

3. The authors concluded that UVR8-mediated sequestration of COP1 in UV-B reduces the stability of PIF5, providing a mechanism for rapid reduction of PIF5 abundance in sunlight. More analysis, such as the effect of loss-of-function cop1 mutation (e.g. cop1-4) should be examined in high R:FR (WL) and low R:FR (FR) conditions supplemented with or without UV-B. Further more, the phenotype of PIF5-overexpressing lines in cop1 in high R:FR (WL) and low R:FR (FR) conditions supplemented with or without UV-B could be examined as well. A recent study that the COP1-SPA complex is necessary for the red light-induced degradation of PIF5 by using PIF5-Myc and PIF5-Myc/cop1-4 has been reported (Pham et al., Plant Journal, 2018) and can be a good example.

We agree that a western blot, showing native PIF5 abundance in WT and *cop1-4* in high R:FR ± UV-B should be included, in addition to low R:FR. This has been included as Figure 5 e, f. These experiments show that in the absence of COP1, PIF5 proteins levels are virtually undetectable in both high and low R:FR. Further decreases following UV-B treatment therefore cannot be observed. We have investigated *PIF5* transcript levels in *cop1* mutants (see response 5 to reviewer 1). Our data are entirely consistent with the model that COP1 is required for PIF5 protein stability. We have also included hypocotyl data from WT and *cop1* seedlings grown in high and low R:FR ± UV-B in Figure S8. These data support previously-published observations that *cop1* mutants are extremely short and unable to perform shade avoidance. As such, *cop1* mutants display no further hypocotyl inhibition upon UV-B treatment. We propose that these phenotypes result, in part, from a severe depletion of PIF5 protein.

Although transgenic plants expressing tagged *PIF5* in the *cop1-4* background have recently been published (Pham et al. 2018), we think that analyses of native protein are more informative for this experiment. We do not think that inclusion of these lines would change our conclusions.

4. COP1 has been shown to physically interact with PIF5 and promotes the ubiquitination and degradation of PIF5 in red light (Pham et al., Plant Journal, 2018). In this study, Fig. 5c shows that COP1 associates with PIF5 in high R:FR. Does COP1 also promote PIF5 degradation in high R:FR? If so, based on the model proposed in this study, UV-B induced UVR8 sequestration of COP1 will inhibit COP1 mediated PIF5 degradation in high R:FR+UV-B. That is, PIF5 will be stabilized in high R:FR+UV-

B. However, Fig. 2 shows PIF5 is rapidly degraded in high R:FR+UV-B. This point require close attention.

The reviewer raises a key point here. In a recently published study by Pham and colleagues (2018, Plant Journal), COP1 is shown to promote the stability of PIF5 in dark-grown seedlings but promote degradation of PIF5 following transfer to red light. Our data show that COP1 binds to and stabilises PIF5 in high AND low R:FR. This is consistent with data from the Pham et al study using light-grown de-etiolated seedlings, in which *cop1* mutants show depleted PIF5 protein in the light (Pham *et al* 2018- Figure 6c). Our model proposes that UVR8-mediated sequestration of COP1 in UV-B results in the destabilisation and degradation of PIF5 in both conditions, although differences are greater in low R:FR, where PIF5 levels are highest. Western blots, showing reduced levels of native PIF5 in *cop1-4* in high and low R:FR are shown in Figure 5e,f. Observed differences in the role of COP1 in regulating PIF5 stability likely represent differences between analysis of dark-grown seedlings transferred to light (ie- de-etiolation) and analyses of fully de-etiolated seedlings. The text has been modified to incorporate this point.

5. For phenotypic analysis in Fig.1, the seedlings were grown in high R:FR for 3 days before UV-B treatment. But for biochemical analysis in Figs. 2-5, the seedlings were grown in 16 h light/8h dark cycles for 10 days before UV-B treatment. Why different conditions were used?

PIF5 protein levels oscillate in diurnal cycles with highest levels recorded at dawn. Biochemical analyses of PIF5 protein were therefore performed in light/dark cycles to ensure that UV-B treatments were given during the period of maximum PIF5 protein abundance. 10 day-old-seedling were used to ensure sufficient material for protein extraction. Hypocotyl length experiments were performed in continuous light to maximise responses and enable the comparison with previous work (Hayes et al. 2014). We have, however, now included additional hypocotyl elongation data from experiments performed in 16 h light/8h dark cycles. These show similar trends to continuous light (Figure S3). Text has been modified accordingly.

6. “Immunoprecipitated PHYB was a lower molecular weight, suggesting that phyB-PIF5 interaction results in modification of the PIF5 protein (Fig. 5a).” PHYB of lower molecular weight led to PIF5 modification? It is not clear how the conclusion is made. Please clarify it. Since PHYB constitutively interacts with PIF5 (Fig. 5a), does PHYB-PIF5 interaction interfere COP1-PIF5 interaction and PIF5 stability? Though PIF5 does not interact with UVR8 as reported (Hayes et al., PNAS, 2014), does UVR8 affect the physical interaction between COP1 and PIF5.

This was a typo and should have read ‘modification of the PHYB protein’. We apologise for the confusion. PHYB was used in this experiment as a positive control for the PIF5 IP. The suggestion that phyB and COP1 may compete for PIF5 binding is interesting and was also raised by reviewer 4. We thank both for their helpful input and have modified the text to incorporate this suggestion.

7. PHYB has been shown to interact with the APB domain of PIF5 in vitro and promote its degradation in red light. In this study, the authors reveal that PHYB constitutively interacts with PIF5, and that the N-terminal ABP domain of PIF5 is required for UV-B-mediated degradation. However, it has been reported that UV-B-mediated inhibition of low R:FR-induced hypocotyl elongation occurs independent of phytochrome (Hayes et al., PNAS, 2014). Are those results consistent with each other?

We are not sure what the reviewer means by constitutive PHYB-PIF5 interaction. We have shown interaction in both the presence and absence of UV-B. Interestingly, we observed similar levels of PHYB in both treatments, despite considerably lower PIF5 levels in UV-B-treated plants, consistent with the competition model described above. Observations of partial hypocotyl growth inhibition by UV-B in *phyABCDE* mutants is consistent with our model. One possibility is that COP1 binding promotes PIF5 stability via additional mechanisms other than simply outcompeting phyB. Indeed, we see significant UV-B-mediated PIF5 degradation in low R:FR where there is a low level of Pfr. In our previous study (Hayes et al. 2014), we showed that UV-B could inhibit PIF activity, via increased DELLA stability, in addition to promoting PIF degradation. We also observed some degree of UV-B-mediated hypocotyl inhibition independently of UVR8. All these mechanisms could inhibit hypocotyl elongation in the *phyABCDE* mutant. We have added some more detail to the discussion to clarify this issue.

8. The immunoblots in Figs. 2-5 did not include corresponding WT or mutant lines as negative control. In Fig. 4b, the immunoblot using ubiquitin blot reflected the ubiquitination of total proteins, but did not reflect the ubiquitination level of PIF5-HA.

Control blots showing an absence of HA signal in WT and an absence of native PIF5 in *pif5*-deficient mutants are provided in Supplementary Figure 6.

We agree with the reviewer that western analysis of PIF5-immunoprecipitates with a ubiquitin antibody would be useful. We have therefore included this experiment in Figure 4c and modified the text accordingly. This experiment was also requested by reviewers 1 and 3.

Reviewer #3 (Remarks to the Author):

This ms reported a study of the molecular mechanism controlling PIF5 degradation in response to UV-B under low R/FR condition. The authors showed that UV-B induced PIF5 protein degradation was mediated by UVR8 photoreceptor through ubiquitin-proteasome pathway, and the N-terminus phytochrome-interacting domain of PIF5 was needed for its degradation. They further proved that PIF5 interacted with COP1 in low R/FR shade-grown plants treated with UV-B to inhibit PIF5 degradation. Given that COP1 is a known partner of UV-B-photoexcited UVR8 monomer, the authors argued that UVR8 mediates PIF5 degradation in response to UV-B light, by sequestration the PIF5 stabilizer: COP1. This seems an interesting hypothesis because it seems distinct with the previously reported stabilizing effects of COP1 on other PIF proteins. The manuscript is well written that addresses a question of general interest in plant photobiology. The data presented are based on some well-designed experiments which pertinently support the conclusion.

Specific comments

1. It is not clear whether UV-B affects COP1-PIF5 interaction. The authors may want to address this issue by a Fig. 5c-related experiment in the presence or absence of UV-B light.

We agree with the reviewer that immunoprecipitation experiments showing COP1-PIF5 interaction with and without UV-B treatment would be helpful. This has now been included in Figure 5c. Reduced levels of PIF5-COP1 complex were observed following UV-B treatment, consistent with our model. Text has been modified accordingly. This experiment was also suggested by reviewer 1.

2. Line 171, “Immunoprecipitated PHYB was a lower molecular weight, suggesting that phyB-PIF5 interaction results in modification of the PIF5 protein (Fig. 5a)”. This sentence does not seem to make much sense. Because the blot shown in Fig. 5a seem to indicate that PIF5 pulled down both slow and fast migrating phyB bands, it may be an artifact of phyB degradation due to the prolonged IP reaction. Regardless, why does this observation argued for “phyB-PIF5 interaction results in modification of the PIF5 protein”? Third, there seems no different effects of UV-B, then how is this experiment associated with the main thesis of this ms?

Apologies for this confusion. This was a typo and should have read ‘modification of the PHYB protein’. PHYB was used in this experiment as a positive control for the PIF5 IP.

3. The authors seem to use Fig. 4b to show the effects of MG132. It may be interesting to show ubiquitylation of the IP products of PIF5.

We agree with the reviewer that western analysis of PIF5-immunoprecipitates with a ubiquitin antibody would be useful. We have therefore included this experiment in Figure 4c and modified the text accordingly. This experiment was also requested by reviewers 1, 2 and 3.

Reviewer #4 (Remarks to the Author):

This manuscript uncovers a novel mechanism by which plants integrate UVB light signals into the complex network they use to adapt their growth and development to a constantly fluctuating light environment. It is becoming increasingly clear that multiple signals converge in plant cells through the small PIF transcription factor family, to regulate a target transcriptional network that transduces these signals into adaptational phenotypic responses. This work from Sharma et al. adds an important new dimension to that picture by showing how the UVB receptor, UVR8, exploits a non-canonical function of an E3 ubiquitin ligase, COP1, to rapidly regulate the abundance of the PIF5 transcription factor in response to UVB light levels.

My only comment is, that it seems to me that COP1 could be acting antagonistically (in dynamic fashion) to the established phyPfr-induced phosphorylation-ubiquitination-degradation mechanism, by interfering with the binding of phy, kinases and/or ligases to PIF5. The UVB/UVR8-COP1 sequestration would then represent a rapid mechanism of modulating the rate of the existing phy/kinase/ligase-driven degradation (and therefore steady-state levels) of PIF5 across the fluctuating light/dark/shade/sunfleck/etc light environment. I would like to see some discussion of how these UVB signals might be integrated into this known framework of PIF-abundance regulation.

We agree and thank the reviewer for this useful suggestion. We have expanded the discussion to include these ideas. We would, however, like to point out that the rapid UV-B-mediated degradation of PIF5 in low R:FR (ie. conditions of low Pfr) suggests that COP1 may act to stabilise PIF5 through additional mechanisms, other than just simply antagonising PhyB Pfr binding.

Some editorial points:

1. Fig. 1b, right panel: There seems to be an extra ‘dot’ there.

We think that the reviewer may be referring to asterisks above data points, highlighting statistically significant differences. We have now increased their size to improve clarity.

2. Fig. 4 legend: Need to define 'FR' here.

This has been corrected.

3. Fig. 5d: Need to indicate that these panels are replicate experiments (if that's what they are).

This figure has been modified and only one repeat is now shown.

4.Lines 142-4: Should read something like: "Blots were also probed with an anti Ubiquitin antibody to confirm that the MG132 had fully infiltrated into the tissue and enhanced accumulation of ubiquitinated proteins (Fig. 4b)".

This has been corrected.

5. Line 157: show/suggest? Need to choose.

This has been corrected.

6. Line 172: Apparent typo. Should be "suggests modification of the phyB protein".

This has been corrected.

Reviewers' comments:

Reviewer #1 (Remarks to the Author):

The revised version of this study is much improved. The quality of data can be improved to meet the standard of Nature Communications as suggested below:

1. Figure 5c, as the COP1 band in the IP fractions looks faint, it is difficult to conclude that UV-B reduces the abundance of PIF5-COP1. The result of WT in the IP fraction (like input fraction) should be shown to exclude the possibility of unspecific binding. To compare the difference of co-immunoprecipitated COP1, three replicates are suggested to present for quantification (like Figure 3a and d).

2. As for the evidence showing the specific disruption of the UVR8-COP1 interaction makes PIF5 stable, the authors think it impossible to study disruption of this process, without affecting a significant number of UV-B responses. It has been reported that some mutation of UVR8 tryptophans (such as UVR8 W92A, W94A) reduces UVR8 interaction with COP1 but has no apparent effect on UVR8 function (O'Hara and Jenkins, 2012, Plant Cell). The material was generated by the co-author Prof. Jenkins's lab, and may help give an answer.

Reviewer #2 (Remarks to the Author):

The revised manuscript has addressed most of my questions. My only concern is vague COP1 bands in IP samples (Figure 5c and 5d). I suggest the authors claim a clear difference using quantification (IP versus input). IP result of the WT control should be presented in Figure 5c.

Reviewer #3 (Remarks to the Author):

This reviewer is satisfied with the authors' responses
The revised ms is significantly improved

NCOMMS-18-36124A: Response to reviewer comments

Reviewer #1 (Remarks to the Author):

The revised version of this study is much improved. The quality of data can be improved to meet the standard of Nature Communications as suggested below:

1. Figure 5c, as the COP1 band in the IP fractions looks faint, it is difficult to conclude that UV-B reduces the abundance of PIF5-COP1. The result of WT in the IP fraction (like input fraction) should be shown to exclude the possibility of unspecific binding. To compare the difference of co-immunoprecipitated COP1, three replicates are suggested to present for quantification (like Figure 3a and d).

We thank the reviewer for their positive appraisal of our revised manuscript and helpful suggestions. We acknowledge that COP1 bands in IP fractions are relatively faint. This results from the low level of COP1 protein in these fractions. We have optimised the procedure as best as we can and have now quantified average IP/input ratios for PIF5-HA and COP1 across multiple biological replicates. These data are shown in Figure 5e, f. We have included a blot with a WT IP control for PIF5-HA and COP1 in Supplementary Figure 6c.

2. As for the evidence showing the specific disruption of the UVR8-COP1 interaction makes PIF5 stable, the authors think it impossible to study disruption of this process, without affecting a significant number of UV-B responses. It has been reported that some mutation of UVR8 tryptophans (such as UVR8 W92A, W94A) reduces UVR8 interaction with COP1 but has no apparent effect on UVR8 function (O'Hara and Jenkins, 2012, Plant Cell). The material was generated by the co-author Prof. Jenkins's lab, and may help give an answer.

The reviewer is likely referring to a figure in the cited paper that shows interaction of the UVR8W92A, W94A mutant with COP1 in yeast; there appears to be a reduced interaction but (i) there is still a clear interaction and (ii) this is in yeast. The data for the UVR8W92A, W94A-COP1 interaction in plants were not shown in the paper but are shown below [redacted] We are not aware of any *uvr8* mutant that retains function when the COP1 interaction is disrupted. If there was such a mutant, we would certainly have used it in this study.

[redacted]

Reviewer #2 (Remarks to the Author):

The revised manuscript has addressed most of my questions. My only concern is vague COP1 bands in IP samples (Figure 5c and 5d). I suggest the authors claim a clear difference using quantification (IP versus input). IP result of the WT control should be presented in Figure 5c.

We thank the reviewer for their helpful suggestions. These analyses have now been performed and are presented in Figure 5 e, f. We have included a blot with a WT IP control for PIF5-HA and COP1 in Supplementary Figure 6c.